# The all-particle energy spectrum of cosmic rays from 10 TeV to 1 PeV measured with HAWC

Jorge A. Morales-Soto[⋆] and Juan C. Arteaga-Velázquez[†]
on behalf of the HAWC Collaboration

Instituto de Física y Matemáticas, Universidad Michoacana de San Nicolás de Hidalgo

⋆ jorge.morales@umich.mx , † juan.arteaga@umich.mx

*21st International Symposium on Very High Energy Cosmic Ray Interactions
(ISVHECRI 2022)
Online, 23-28 May 2022*

## Abstract

The HAWC observatory is an air-shower detector, which is designed to study both astrophysical gamma-rays in the TeV region and galactic cosmic rays in the energy interval from 1 TeV to 1 PeV. This energy regime is interesting for cosmic ray research, since indirect observations overlap with direct measurements, which offers the opportunity for cross calibration and studies of experimental systematic errors in both techniques. One quantity that could help for this purpose is the all-particle energy spectrum of cosmic rays. In this work, we present an update of HAWC measurements on the total cosmic-ray energy spectrum between 10 TeV and 1 PeV. The spectrum was obtained from an unfolding analysis of almost two years of HAWC's data, which was collected from January, 2018 to December, 2019. For the energy estimation, we employed the high-energy hadronic interaction model QGSJET-II-04. As in a previous work of HAWC, published in 2017, we observed the presence of a knee-like feature in the region of tens of TeV.

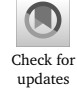
## 1 Introduction

A challenge for direct and indirect cosmic ray experiments has been the precise measurement of the total spectrum of cosmic rays in the energy interval from 10 TeV to 1 PeV. In this energy region, early measurements from direct experiments, such as ATIC-02 [1] and CREAM [2], exhibit low statistics, while previous measurements from indirect air shower observables, such

---

A complete list of HAWC's authors is available at: https://www.hawc-observatory.org/collaboration/.

as ARGO [3] and TIBET [4], are constrained to the energy region just below 1 PeV. The development of new technology and the increasing interest in studying this energy region, have encouraged the construction of a new generation of cosmic ray experiments (e.g. NUCLEON [5], HAWC [6], and LHAASO [7]) to make precise measurements of the energy spectrum in the multi-TeV energy range.

The High Altitude Water Cherenkov (HAWC) observatory is located at 4100 m a.s.l. at the Sierra Negra Volcano in Puebla, Mexico, and is composed of a dense air shower array of 1,200 photomultipliers (PMTs) installed in 300 water Cherenkov tanks. The Cherenkov detectors contain a total of 60 ML of water and are distributed over a flat surface of $22,000\,m^2$.

One of the main science goals of the HAWC collaboration is to study cosmic rays in the TeV regime. The HAWC collaboration published first results on the all-particle cosmic-ray energy spectrum between 10 and 500 TeV in 2017 using 8 months of data [6]. In this work, the collaboration reported the existence of a break in the energy spectrum, at $(45.7 \pm 1.1)$ TeV, which was also reported by NUCLEON [5].

With more available data and improved simulations on the performance of the detector, the present study provides an updated HAWC result on the all-particle energy spectrum of cosmic rays between 10 TeV and 1 PeV, thus improving the statistical and systematic uncertainties in the spectrum, increasing the effective time of the data, and extending the previous HAWC measurements up to 1 PeV. The analysis is based on the Bayes' unfolding method [8].

## 2 HAWC simulation, experimental data and quality cuts

The production and development of air shower events were simulated using the CORSIKA (v760) code [9], where the hadronic interactions are treated by FLUKA [10] for hadronic energies < 80 GeV, and by QGSJET-II-04 [11] for higher energies. The passage of the incoming secondary particles through the HAWC's detectors was simulated with GEANT-4 [12]. According to [6, 13] a total of eight different primary nuclei (H, He, C, O, Ne, Mg, Si and Fe) were simulated. The primary particles were generated from an $E^{-2}$ differential energy spectrum and for arrival directions with zenith angles < 65°. The simulations were weighted according to their mass and energy to simulate elemental cosmic-ray spectra according to a composition model based on a fit to experimental measurements at TeV energies [2, 14, 15]. On the other hand, the measured data employed in the analysis were collected during the period from January, 2018 to December 2019 and spans an effective time of observation of 1.9 years. Both data and MC simulations were reconstructed according to the algorithms described in [16]. MC simulations were used to study systematic errors and for the estimation of the effective area and the response matrix in the unfolding procedure of the energy spectrum. To determine the energy of primary cosmic rays, we performed a maximum log-likelihood estimation for which the measured lateral distribution of the event is compared with probability tables of such distributions computed for different energies and zenith angle intervals using QGSJET-II-04 and protons as primaries since these nuclei are the most abundant particles in the energy range under study. For more details about the primary energy reconstruction see [6, 13]. To diminish the effect of systematic uncertainties due to the reconstruction of the core position and the arrival direction of air showers, several selection cuts were applied to the data. The selection criteria were derived following studies with Monte Carlo (MC) simulations. For the present analysis, the events must have successfully passed the event reconstruction procedure described in [16], shower axes must have zenith angles $\theta \le 35°$, and should have activated a minimum of 60 photomultipliers (PMTs) within a radius of 40 m from the core of the event. In addition, the reconstructed shower cores are required to be mainly inside HAWC's area, and to have more than 30% of the active PMTs with signals. Also, the selected data was restricted

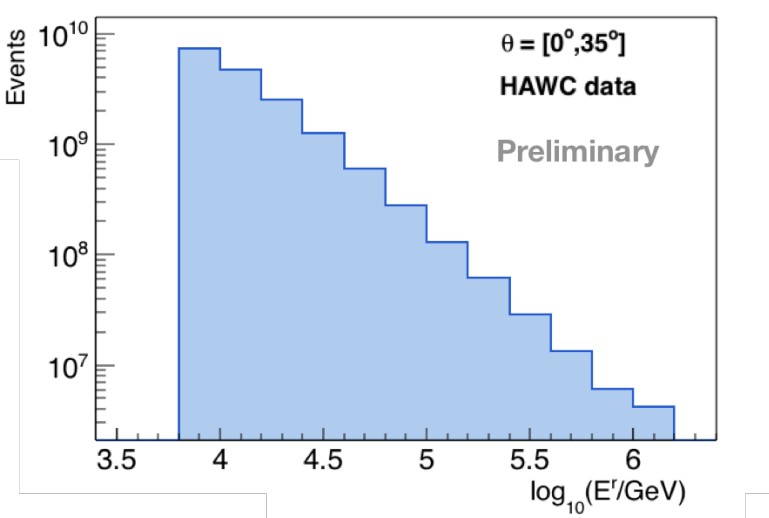

Figure 1: Raw energy distribution, $N(E^r)$, of the selected HAWC data. The quality cuts described in section 2 were applied.

to the reconstructed energy interval $\log_{10}(\text{E/GeV}) = [3.8, 6.2]$. From studies with MC simulations the energy, angular and shower core resolutions at E = 10 TeV are 52%, 0.5°, and 13.1 m, respectively, while at E = 1 PeV the corresponding values are 37%, 0.6° and 15.4 m. Once the quality cuts were applied to the experimental data, a total of $1.3 \times 10^{12}$ shower events are remaining in the selected data set.

## 3 Reconstruction of the energy spectrum

To begin with, we obtained the raw energy distribution, $N(E^r)$, of the data based on the event selection mentioned in section 2. This distribution is just the event histogram for the reconstructed primary energy, $E^r$, without any correction for migration effects. The spectrum is built from the selected measurements using a bin size of $\Delta \log_{10}(E^r/\text{GeV}) = 0.2$ (see Fig. 1), and it must be corrected for migration effects. For this aim, we use the Bayes unfolding procedure [8]. The response matrix, $P(E^r|E)$, is derived from MC simulations using our cosmic-ray composition model (see Fig. 2, left). Here, $E$ corresponds to the true primary energy.

Then, from our MC simulations (see section 2), we computed the effective area (c.f. Fig. 2, right) by means of the formula

$$A_{\text{eff}}(E) = A_{\text{thrown}} \cdot \epsilon(E). \tag{1}$$

$A_{\text{thrown}}$ stands for the area of a circular region with radius 1 km over which the simulated events were thrown multiplied by a geometrical factor due to the flat geometry of the array and $\epsilon(E)$ is the efficiency for detecting a shower event with energy $E$, which is estimated with MC simulations and our cosmic-ray composition model. For further details on how this procedure is done see [6].

Finally, the energy spectrum is estimated using the formula

$$\phi(E) = \frac{N(E)}{\Delta E \, T \, \Delta\Omega \, A_{\text{eff}}}, \tag{2}$$

where $N(E)$ is the unfolded spectrum and $\Delta E$ represents the size of the energy bin centered

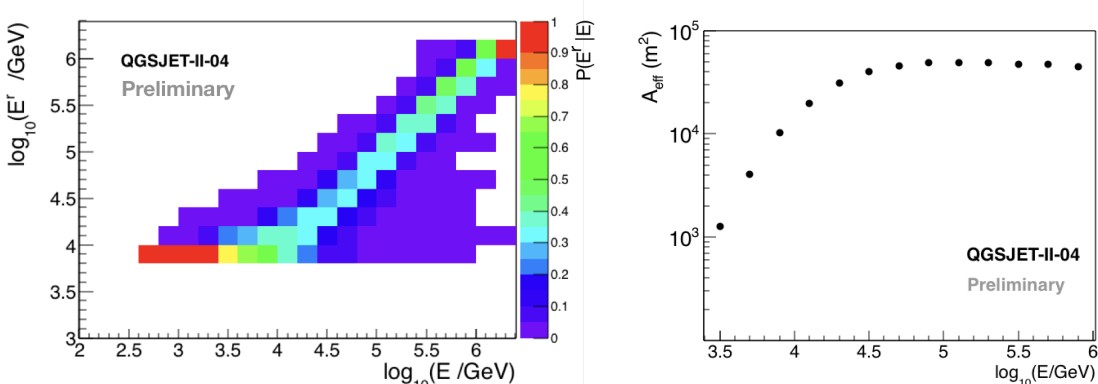

Figure 2: *Left panel*: Response matrix $P(E^r|E)$ estimated with QGSJET-II-04 and our nominal cosmic-ray composition model. *Right panel*: Effective area for the reconstruction of the energy spectrum of cosmic rays as a function of the true primary energy E. It was calculated with our MC simulations, using QGSJET-II-04.

at E. The total effective time of observation corresponds to $T = 703$ days. The solid-angle interval is $\Delta\Omega = 1.14\,sr$.

## 4  Results

The unfolded energy spectrum obtained from this analysis is shown in Fig. 3, left, where the error band represents the systematic uncertainty. The error bars show the statistical errors, which include the uncertainties due to the limited statistics from both the data and the response matrix. At an energy close to E = 1 PeV, the statistical uncertainty is $\pm$ 0.01%. At the same energy, the systematic error is found between -3.7% and +9.6%. At energies E = 10 TeV, the statistical uncertainty is $\pm$ 0.01%, and the systematic uncertainty varies from -6.3% and 9.5%. The sources of systematic errors that were included in this estimation are uncertainties in the relative abundances of cosmic rays, and the effective area, the quantum efficiency/resolution of the PMT's [16], the charge resolution and late light simulation of the PMT's [16], the uncertainty of the minimum energy threshold of the PMT's [16] and the unfolding technique (using Gold's unfolding algorithm [17] in the reconstruction procedure to evaluate variations in the main result, and studying also the dependence with the prior and the smoothing algorithm). From Fig. 3, we observe that the spectrum breaks at TeV energies. In order to find out whether a spectrum with a break is preferred by the data over a simple power-law behaviour, we applied a statistical analysis. First, we fitted the total spectrum with a $\chi^2$ minimization procedure, described in [18], using a power-law formula

$$\Phi(E) = \Phi_0 E^{\gamma_1}, \tag{3}$$

where $\Phi_0$ is used as a normalization parameter, and $\gamma_1$ is the spectral index. The results from the fit are $\Phi_0 = 10^{4.48\pm0.01}$ m$^{-2}$ s$^{-1}$ sr$^{-1}$ GeV$^{-1}$ and $\gamma_1 = -2.649 \pm 0.001$ with $\chi_0^2 = 406.36$ for 8 degrees of freedom. Then, we apply a $\chi^2$ fit to the measured spectrum with a broken power-law function

$$\Phi(E) = \Phi_0 E^{\gamma_1}\left[1 + \left(\frac{E}{E_0}\right)^{\epsilon}\right]^{(\gamma_2-\gamma_1)/\epsilon}. \tag{4}$$

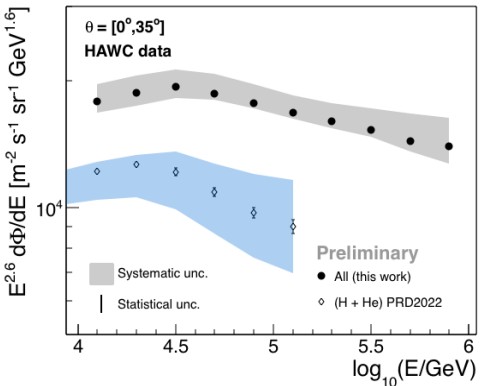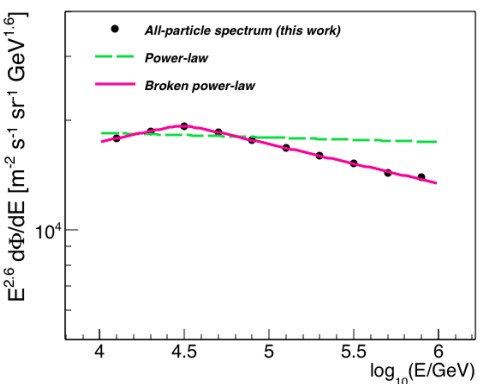

Figure 3: *Left panel*: The total energy spectrum of cosmic rays measured with HAWC (black circles) compared with the spectrum of the H+He group of cosmic rays from [13] (open diamonds). Error bands and vertical error bars represent systematic and statistical uncertainties. *Right panel*: Results of the fits to the HAWC spectrum with a power-law formula (dashed line) and a broken power-law expression (continuous line).

The fit yields $\Phi_0 = 10^{3.84 \pm 0.041}$ m$^{-2}$ s$^{-1}$ sr$^{-1}$ GeV$^{-1}$, $\gamma_1 = -2.5 \pm 0.009$, $\gamma_2 = -2.7 \pm 0.004$, $\epsilon = 9.9 \pm 1.8$, and $E_0 = (30.84^{+1.83}_{-1.72})$ TeV with $\chi_1^2 = 0.21$ for 5 degrees of freedom. The results of the above fits can be seen in Fig. 3, right. As a next step, the test statistic, $TS = \Delta\chi^2 = \chi_0^2 - \chi_1^2$, is employed to find out which hypothesis best describes the data. For our result we have that $TS_{\text{obs}} = 406.15$. For the test, we generate toy MC spectra with correlated data points using our statistical covariance matrix and the results of the fit for the power-law model [18], then, we repeated, for each MC spectrum, the fits with eqs. (3) and (4). From these results, we calculated the distribution of the TS under the hypothesis that the spectrum follows a power law. From the TS distribution, it was found that the *p-value* for $TS_{\text{obs}}$ is $p \leq 8 \times 10^{-6}$, giving the broken power-law scenario a significance of at least $4.3\sigma$.

In Fig. 4, the all-particle cosmic ray energy spectrum obtained here is compared with the results from other cosmic ray experiments. The measurements are from the satellites ATIC-02 [1] and NUCLEON [5], and from the indirect cosmic ray experiments ARGO-YBJ [3], ICETOP [19], KASCADE [20, 21], TAIGA-HiSCORE [22], TIBET [4] and TUNKA-133 [23].

## 5 Discussion

According to the present analysis, the all-particle energy spectrum of cosmic rays is not described by a power-law formula in the energy range from 10 TeV to 1 PeV as shown also by NUCLEON [5] and by a previous study with HAWC [6]. From Fig. 4 our result is in agreement within systematic uncertainties with the data from NUCLEON. There is also an agreement with ATIC-02 [1] data at energies close to 10 TeV and with TAIGA-HiSCORE [22] at around 1 PeV. The HAWC result, however, is larger than ARGO-YBJ, TIBET and ICETOP measurements at high energies. We must point out that in this study, we have reduced the systematic uncertainties of the HAWC energy spectrum in comparison with the analysis of [6] thanks to the recent improvements in the PMT modeling. As a reference, at energies of E = 100 TeV, the systematic uncertainty was reduced from -24.8%/+26.4% to -3.7%/+9.7%, with respect to the systematic uncertainties reported in [6].

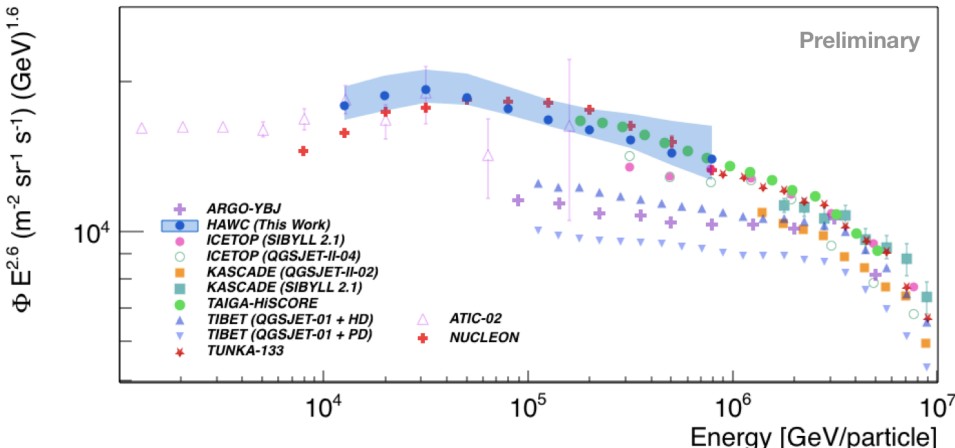

Figure 4: Comparison of the HAWC all-particle energy spectrum of cosmic rays (black circles, this work) and other measurements of the spectrum from direct (ATIC-02 (violet open triangles) [1], NUCLEON (red crosses) [5]) and indirect (ARGO-YBJ (violet crosses) [3], ICETOP (pink dots and green open circles) [19], KASCADE (orange squares and green squares) [20, 21], TAIGA-HiSCORE (green circles) [22], TIBET (upward blue triangles and downward blue triangles) [4] and TUNKA-133 (red stars) [23]) cosmic-ray experiments.

In Fig. 3, we have also compared the HAWC results with recent measurements of the observatory on the H+He energy spectrum of cosmic rays in the TeV region. We observe that the spectrum of protons plus helium nuclei also shows a softening, but at an energy of 24 TeV, i.e., at a lower energy than the one for the position of the knee-like feature in the all-particle energy spectrum. This difference may be due to the influence of the heavy component in the total spectrum of cosmic rays.

## 6  Conclusions

An improved analysis of HAWC data on air showers induced by TeV cosmic rays has allowed to estimate the all-particle energy spectrum from 10 TeV to 1 PeV, where direct and indirect measurements of cosmic rays overlap. A comparison of HAWC data with direct measurements of the NUCLEON space observatory in the 10 TeV - 1 PeV energy range shows that the results of both experiments on the all-particle spectrum of cosmic rays are in agreement within systematic uncertainties. HAWC measurements show a softening in the all-particle spectrum at around 31 TeV with a statistical significance of at least $4.3\sigma$, just at an energy above the position of the knee-like structure that is observed in the HAWC energy spectrum of H+He cosmic rays after the implementation of a statistical analysis.

## Acknowledgements

The main list of acknowledgements can be found under the following link: https://www.hawc-observatory.org/collaboration.

**Funding information** In addition, J.A.M.S. and J.C.A.V. also want to thank the partial support from CONACYT grant A1-S-46288, and the Coordinación de la Investigación Científica de la Universidad Michoacana.

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
