# Peer review of "The all-particle energy spectrum of cosmic rays from 10 TeV to 1 PeV measured with HAWC"

_SciPost Physics Proceedings, doi:SciPost Phys. Proc. 13, 039 (2023)_

## Round 2 · Referee Report · Anonymous · 2022-11-17

Report

This paper describes an update of the all-particle cosmic ray spectrum between 10 TeV and 1 PeV obtained from almost 2 years of data from the HAWC observatory. This work extends a previous analysis by HAWC to higher energies and uses a larger event sample. The paper reports the observation of a softening of the cosmic ray spectrum at around 30 TeV which is favored over a simple power law assumption at the 4.3 sigma level. As this result confirms previous observations by the HAWC and NUCLEON collaborations, and it is also in agreement with other measurements at around 1 PeV within uncertainties, it is of large interest for the cosmic ray community. The paper is very well organized and written clearly, the results are presented in a very clear manner, and I highly recommend the publication of this paper in SciPost Physics Proceedings. I only have a few comments and recommendations for consideration (mainly minor remarks on wording) before publication which are listed in the following under "requested changes".

Requested changes

1- Abstract: " This energy regime is quite interesting..." I recommend to remove the word "quite".

2- p1, Introduction, first sentence: "One of the most challenging tasks..." Certainly, this is an important task for cosmic ray experiments, however, I think the wording is too strong as there are more challenging tasks, such as measurements of the cosmic ray mass composition or anisotropy, for example. Thus, I suggest to re-phrase this statement slightly by writing "A challenge for...", or "One of the core tasks...", for example.

3- p1, last line: There is a line break between "22,000" and its unit "m^2". I recommend to try to avoid this line break.

4- p2, 1st paragraph: "In that work,..." -> "In this work,...".

5- p2, 2nd paragraph & last line of the page: "Bayes" -> "Bayes'" (apostrophe added).

6- p2, section 2, 1st paragraph: This is a lengthy first sentence and I believe the commas are not correct. I recommend to split the sentence and write, for example: "The production and development of air shower events were simulated using the CORSIKA (v760) code [9], where the hadronic interactions are treated by FLUKA [10] for hadronic energies < 80 GeV, and by QGSJET-II-04 [11] for higher energies. The passage of the incoming secondary particles through the HAWC’s detectors was simulated with GEANT-4 [12]. "

7- p2, center: "According to [6,13] a total of eight different primary nuclei (H, He, C, O, Ne, Mg, Si and Fe) simulated." A verb is missing here, i.e. "...were simulated.". Also, to my understanding only the proton (H) simulations are used in this work, why are the other primaries mentioned (see also my comment no. 9)?

8- p2, center: "...with zenith angles < 65◦ [6]." I suggest to drop the reference [6] here since it is already mentioned in the previous sentence which is related to this sentence.

9- "...using QGSJET-II-04 and protons as primary nuclei [6]. " Why are only proton simulations used when other primaries have been simulated (see also my previous comment no. 7)? I assume the reasoning is that in this energy range the flux is dominated by protons, however, I believe this should be explained in the text. Also, I think reference [6] can be dropped in this case.

10- p2, end of section 2: You introduce the abbreviation "PMT" in the text, however, the ord "photomultiplier" is used a few lines before already. I recommend to introduce the abbreviation upon first mentioning of "photomultipliers".

11- p2, end of section 2: "In addition, the shower data were asked to have reconstructed shower cores mainly inside HAWC’s area, ..." Please consider a different wording since data can not be asked... I recommend to write, for example: "In addition, the shower data were asked to have reconstructed shower cores mainly inside HAWC’s area, ..."In addition, the reconstructed shower cores are required to be mainly inside HAWC's area, ...".

12- p2, end of section 2: You state the energy and angular resolution at the highest energy, i.e. 1 PeV, however, I believe the resolutions at lower energies are also of interest for the reader and important to state, i.e. what are the corresponding resolutions at 10 TeV?

13- p2, last sentence of section 2: "..., we are left with a total of..." I suggest a rewording here, for example: "..., a total of 1.3x10^12 shower events are remaining...".

14- p2, first sentence of section 3: "To begin with, we get the raw energy distribution, N(E^r), from the data." What is the "raw energy distribution"? What is N, what is E^r? I believe this needs some more explanation.

15- p3, top: "P(E^r|E)" what is E ?

16- p3, Fig. 1 and Fig. 2: I highly recommend to place Fig. 2 on top, Fig. 3 on the bottom of the page as currently the text is split by the figures which makes it hard to read. Please re-consider the figure placement here.

17- p3, center: "In Eq. (1), A_thrown stands for the area of..." I suggest to drop the first sub-sentence and just write " A_thrown is the area of..." (you don't need the reference to Eq. (1) as the sentence is directly following). In the same sentence, I assume that "R_thrown" is the radius of the disk that is used to throw the showers. I recommend to explain this procedure better in the text.

18- p4, last sentence of section 3: This is also a quite lengthy sentence which I suggest to split, for example as "...where N(E) is the unfolded spectrum and ∆E represents the size of the energy bin. The total effective time..."

19- p4, first sentence of section 4: Also, a bit lengthy sentence which I suggest to split, for example "...the systematic uncertainty. The error bars show the statistical errors..."

20- p4, beginning of section 4: "At an energy close to E = 1 PeV,..." What about the resolutions at lower energies? I think it's worth to comment on this (see also my previous comment no. 12).

21- p4, center: "...that the spectrum shows a cut at TeV energies." -> "...that the spectrum breaks at TeV energies." (or similar)

22- p4, paragraph after Eq. (4): The "TS =..." formula in the text extends over the text margin, please fix.

23- p4, paragraph after Eq. (4): In "TS_obs" the subscript "_obs" should be in text mode, i.e. please use \mathrm{} or similar, just as you do in "A_eff" or "R_thrown" before.

24- p4, 2nd-to-last line: "...from the indirect cosmic ray experiments..." Earlier in the text you use "cosmic-ray experiments", please be consistent throughout the text.

25- Caption of Fig. 3: All text is in italic font, however, I think you only want the "Left panel" and "Right panel" ("panel" added for consistency, see previous figures) to be italic.

26- Fig. 3 and Fig. 4: Why is the uncertainty band so assymmetric? I think you should comment on this in the text, in particular because this seems to have changed w.r.t. to the previous HAWC result. The same comment hold for the first paragraph on page 6, i.e. "...was reduced from -24.8%/+26.4% to -3.7%/+9.7%.".

27- 1st sentence of the conclusions: I think you should drop "of this radiation".

28- 2nd sentence of the conclusions: I think you should drop "In this regard".

29- Conclusions: I believe it would be good to state the significance of the broken power-law scenario again as this is an important result of this work.

  • validity: high
  • significance: high
  • originality: high
  • clarity: top
  • formatting: good
  • grammar: good

Author:  Jorge Antonio Morales Soto  on 2023-01-14  [id 3234]

(in reply to Report 1 on 2022-11-17)

we followed the suggested changes and we added a more detailed explanation about the energy calibration method to the text.

---

## Editorial Decision

published